A holistic approach to development of diets for Ballan wrasse (Labrus berggylta) – a new species in aquaculture

Hamre Kristin 1 kha@nifes.no
Nordgreen Andreas 2
Grøtan Espen 3
Breck Olav 3
1 National Institute of Nutrition and Seafood Research (NIFES) , Bergen , Norway
2 Nofima AS , Fyllingsdalen , Norway
3 Marine Harvest Labrus , Bergen , Norway
Conceição Luis
Electronic publication date: 2013 Jul 16
Publication date: 2013
Volume: 1
Electronic Location ID: e99
Received 2013 May 2; Accepted 2013 Jun 17
Copyright: © 2013 Hamre et al.
Copyright year: 2013
Copyright holder: Hamre et al.
License: This is an open access article distributed under the terms of the Creative Commons Attribution License, which permits unrestricted use, distribution, and reproduction in any medium, provided the original author and source are credited.
License URL: https://creativecommons.org/licenses/by/3.0/

Keywords: Ballan wrasse, Nutrient requirements, Broodstock diet, Ongrowing diet

Funding: Norwegian Research Council The Norwegian Seafood Research Fund Marine Harvest Villa Organic AS The work has been funded by the Norwegian Research Council (Project 200523/S40), The Norwegian Seafood Research Fund - FHF (Project 900554), Marine Harvest and Villa Organic AS. Espen Grøtan and Olav Breck are employed by Marine Harvest to work with the development of commercial culture of Ballan wrasse.

==============================
Wild wrasses are used for delousing of farmed salmon but increasing demands have prompted the salmon industry to develop cultures of Ballan wrasse. One of the bottlenecks has been nutrition and feed intake in the juvenile phase, while broodstock nutrition is considered critical for production of viable offspring. The present study was aimed at developing functioning ongrowing and broodstock diets for Ballan wrasse. In juveniles the best lengthwise growth was identified at 65% dietary protein, 12% lipid and 16% carbohydrate. To investigate if the requirements for the other nutrients were covered by the diets developed for the species, the nutrient composition in juveniles (whole body) and broodstock (female gonad) were analyzed and compared to the composition in wild fish. We found that the levels of the lipid soluble Vitamins A, K and D were lower in cultured than in wild fish, however, the requirements for these nutrients in Ballan wrasse are not known. Other candidate nutrients for more in-depth investigation are the bone minerals, zinc, taurine and fatty acids.

Introduction

Salmon lice are currently a major environmental and economic issue in the salmon industry (Bjørn et al., 2012). In some geographical regions, lice have developed resistance to commonly used medications (Horsberg, 2000), which has led to more frequent delousing and an associated increase in treatment costs. The medications used may pose an environmental threat, since some bio-accumulate in the environment and are toxic to wild fauna, such as non target crustaceans, near the aquaculture sites (Horsberg, 2000). In addition, increased levels of lice on farmed salmon can increase the burden of salmon lice on wild salmonids (Bjørn et al., 2012).

An environmental friendly treatment alternative is biological delousing of salmon with cleaner fish. Wild caught wrasse have been used for delousing salmon for at least two decades (Kvenseth, 2011), but the increasing incidence of drug resistant salmon lice has increased the demand for cleaner fish to such an extent that questions have been raised concerning wild catch posing a negative effect on the ecosystems (Mortensen & Karlsbakk, 2012). This is the background for the recent efforts to farm Ballan wrasse (Labrus bergylta).

Commercial juvenile production of Ballan wrasse was established using knowledge and infrastructure from Atlantic cod (Gadus morhua) farming and showed some success within two years, however, numbers are still low and there are many technical and biological challenges. One problem was that fish transferred to weaning and grow-out diets had slow growth and high mortalities (Grøtan, pers. comm.). One reason for this was later shown to be the absence of attractants in the commercial diets used. Addition of shrimp meal to the diets increased feed intake and improved rearing results to an extent (A Nordgreen, E Grøtan and K Hamre, unpublished data; I Opstad, PG Kvenseth, P Jensen and AB Skiftesvik, unpublished data). Furthermore it was hypothesized that commercial fish diets developed for other species such as Atlantic cod may be inappropriate for Ballan wrasse. For instance, Ballan wrasse is a stomachless fish with an intestine length of only 2/3 of its body length (Hamre & Sæle, 2011), and its feeding habits and natural prey selection suggest nutrient requirements and digestive capacity that differ from other commercial marine fish species. Ballan wrasse feed on invertebrates, with species in the classes Gastropoda, Decapoda, Echinodermata and Bivalvia being the most abundant feed items in intestines from wild fish caught at Azores, in the North Atlantic Ocean (Figueiredo et al., 2005). This is similar to the feed selection among other wrasse species according to Lek et al. (2011) and indicates that the natural diet of Ballan wrasse is relatively low in lipids and easily digestible.

The present study is a first approach to screen for nutrient requirements of Ballan wrasse in order to facilitate its domestication. Optimal macronutrient compositions were studied using a three component mixture design, where protein, lipid and carbohydrate were varied systematically within the requirement ranges found for fish in general. Based on previous experience, the results from the macronutrient experiment, and assumptions regarding protein and lipid sources and micronutrient supplementation, a new diet (Labrus, Skretting AS) was formulated for use in the commercial production of wrasse. The whole body nutrient composition of fish groups fed the Labrus Skretting diet, or a moist diet based on Skretting Vitalis (a commercial broodstock diet) blended with shrimp, for at least three months was compared with the whole body nutrient composition of wild fish. A similar approach was used for broodstock that had been fed the moist diet (Vitalis, Skretting AS blended with shrimp) for one year. Female gonads from these fish were analyzed and compared with female gonads from wild fish. Both groups of fish were sampled just before spawning. Assuming that the dietary intake of wild fish fulfills their nutrient requirements and this is reflected in organ nutrient status, large differences in nutrient composition between wild and cultured fish were interpreted as nutritional imbalances in the cultured fish.

Materials and Methods

This study was carried out within the Norwegian animal welfare act guidelines (code 750.000) at Marine Harvest Labrus. As the fish trials were assumed to be nutrition trials based on all available studies up to the date of the trial, no specific permit was required under the guidelines.

Diets

The diets for the macronutrient study were produced by NOFIMA. All dry ingredients for each diet were weighed and carefully mixed to a homogenous blend. Fresh, bone free cod fillets (Gadus morhua) were blended in a standard food processor. The oil, dry ingredients and ethoxiquin (calculated to 25 mg/kg on dry weight basis) were then added while mixing to create a homogenous blend. Diet formulations and analyzed macronutrient compositions are given in Table 1. The feed dough of each diet was fed with a manual press into plastic sausage casings with a diameter of 2 cm. The sausages were heat treated for 12 min at 83°C with 100% humidity in a convection oven (SCC 202, Rational AG, Landsberg am Lech, Germany) to denature the protein. After cooling, the plastic casing was removed and the heat denaturated diets were dried for approximately 24 h at 40°C in a shelf dryer. The water contents of the different diets were analyzed with a Mettler Toledo HG53 Moisture Analyzer during drying to ensure the water content in each diet was similar. Diets were then ground (As200 Basic, Retsch, Düsseldorf, Germany) and sieved to obtain the required particle sizes. Homogenous samples of all diets were analyzed for crude protein (Nx6.25), ash, lipid (Bligh and Dyer) and water content. Diet carbohydrate concentrations were estimated based on subtraction. The feed was packed in closed plastic bags and stored at 4°C until feeding.

Table 1 Diets for the macronutrient experiment (g 100g-1 DW).

A. Basic formulation	
Diet no	1	2	3	4	5	6	7*	8	9	10	11	12	13	
Cod filleta	71.3	63.1	54.9	46.6	62.4	54.3	46.3	37.7	53.7	45.6	37.6	28.9	23.3	
Shrimp mealb	15	15	15	15	15	15	15	15	15	15	15	15	15	
Fish oilc	2.9	11.1	19.3	27.6	3.0	11.1	19	27.5	3	11.1	19.3	27.5	27.4	
Corn Suprexd	4.1	4.1	4.1	4.1	12.9	12.9	12.95	12.96	21.54	21.59	21.7	21.8	27.4	
Wheate	1.8	1.8	1.8	1.8	1.8	1.8	1.8	1.8	1.8	1.8	1.8	1.8	1.8	
Soy lecithinf	3	3	3	3	3	3	3	3	3	3	3	3	3	
Vitamin mixg	0.3	0.3	0.3	0.3	0.3	0.3	0.3	0.3	0.3	0.3	0.3	0.3	0.3	
Mineral mixh	0.55	0.55	0.55	0.55	0.55	0.55	0.55	0.55	0.55	0.55	0.55	0.55	0.55	
Monosodium phosphatei	1	1	1	1	1	1	1	1	1	1	1	1	1	
Carophyll pinkj	0.03	0.03	0.03	0.03	0.03	0.03	0.03	0.03	0.03	0.03	0.03	0.03	0.03	
Yttrium oxide	0.025	0.025	0.025	0.025	0.025	0.025	0.025	0.025	0.025	0.025	0.025	0.025	0.025	
Ethoxiquin	0.0025	0.0025	0.0025	0.0025	0.0025	0.0025	0.0025	0.0025	0.0025	0.0025	0.0025	0.0025	0.0025	
B. Nutrient composition, formulated.	
Diet no	Carbohydrates	Lipid	Protein	Ash	Total (%)	
1	5.00	10.00	77.4	9.2	101.60	
2	5.00	18.33	69.2	8.6	101.13	
3	5.00	26.67	61.3	8.1	101.07	
4	5.00	35.00	53.4	7.5	100.90	
5	12.50	10.00	70	8.7	101.20	
6	12.50	18.33	62	8.1	100.93	
7*	12.50	26.67	54.5	7.6	101.27	
8	12.50	35.00	46.3	7.1	100.90	
9	20.00	10.00	62.9	8.2	101.10	
10	20.00	18.33	55	7.7	101.03	
11	20.00	26.67	47.3	7.1	101.07	
12	20.00	35.00	39	6.6	100.60	
13	25.00	35.00	34.3	6.3	100.60	
C. Nutrient composition, analyzed.	
Diet no	Lipid	Crude protein (N × 6.25)	Ash	Carbohydrates (subtraction)	
1	10.5	80.9	8.0	0.6	
2	19.0	72.1	7.9	1.0	
3	27.0	63.7	7.3	2.0	
4	35.4	55.4	6.9	2.3	
5	10.9	72.1	7.9	9.2	
6	18.8	64.6	7.4	9.2	
7	26.2	57.2	7.0	9.6	
8	34.4	48.5	6.4	10.7	
9	10.1	64.6	7.6	17.7	
10	16.6	59.0	6.6	17.8	
11	26.3	48.7	6.5	18.4	
12	33.2	41.4	6.1	19.4	
13	32.5	36.2	5.8	25.6	
Notes.

a Fresh fillet from wild caught Atlantic cod.

b Shrimp powder (7411), Seagarden AS, Avaldsnes, Norway.

c Norsalmoil®, produced from Capeline. Egersund Sildeoljefabrikk AS, Egersund, Norway.

d Suprex Corn fine, Codrico, Rotterdam, Netherlands.

e Wheat grain (510130), Norgesmøllene AS, Nesttun, Norway.

f Soylecithin GMO powder (20022), Agrosom, Mölln, Germany.

g D3 3000 IE kg-1, E 160 mg kg-1, K3 20 mg kg-1, C 500 mg kg-1, B1 20 mg kg-1, B2 30 mg kg-1, B6 25 mg kg-1, B12 5 µg kg-1, B5 60 mg kg-1, Folic acid 10 mg kg-1, Niacin 200 mg kg-1, Biotin 1 mg kg -1.

h BOLIFOR® MSP, Yara AS, Norway.

i Mn 30 mg kg -1, Mg 750 mg kg-1, Fe 60 mg kg-1, Zn 120 mg kg-1, Cu 6 mg kg-1, K 800 mg kg-1, Se 0.3 mg kg-1.

j Carophyll Pink (10%), DSM, Basel, Switzerland.

* Centre point diets fed to fish in 3 replicate tanks. The other diets were fed to fish in one tank.

The analyzed nutrient compositions of the commercial diets, Labrus and Vitalis (Skretting, Stavanger, Norway), and the moist diet made from Vitalis and shrimp are given in Table 2. The moist diet was made by crushing 100 kg Vitalis cal (Skretting) and mixing in 40 kg boiled, frozen/thawed and minced Greenland shrimp (local supplier), 4.8 kg wheat gluten for binding and 22 L fresh water. The dough was then pelleted in a pelleting machine.

Table 2 Nutrient composition (in dry matter) of the diets fed to Ballan wrasse juveniles and captive broodstock.

	Labrusa	Vitalisb
2010	Vitalis + shrimp
2010	Vitalis + shrimp
2011	NRC (2011)	
A. Macronutrients and taurine (%)	
Dry matter	90	91	52	60	-	
Protein	54	58	62	60	30–60	
Taurine	0.39	na	na	0.37	NR	
Lipid	13	21	18	18	-	
B. Vitamins (mg kg-1)	
Vitamin C	456	736	290	768	50	
Biotin	0.59	0.45	0.44	0.42	0.15–1	
Folic acid	3.8	31	26	28	1.0	
Niacin	156	246	932	384	10–28	
Pantothen	70	77	67	60	10–50	
Vitamin B6	868	27	1528	na	3–6	
Thiamine	12	308	271	155	1.0	
Riboflavin	20	26	23	13	4–7	
Cobalamin	0.21	na	na	0.19	0.015–0.053	
Sum Vitamin A	7.9	6.0	5.6	3.9	0.8	
Vitamin D	0.10	na	na	0.14	0.01–0.04	
Vitamin E	na	511	273	36	50	
Menadione bisulfitec	0.036	0.034	0.010	0.025	NT	
Sum Vitamin Kc	0.14	0.12	0.13	0.11	NT	
Astaxanthin	36	57	46	0	-	
C. Minerals	
Macrominerals (g kg-1)	
Ca	23	na	na	19	-	
Na	16	na	na	10	-	
K	10	na	na	8	-	
Mg	3.3	na	na	2.0	0.4–0.6	
P	20	na	na	13	6–8	
Microminerals (mg kg-1)	
Fe	0.24	na	na	0.32	0.03–0.15	
I	4.4	na	na	4.7	0.6	
Mn	30	45	41	37	2–12	
Cu	21	16	19	24	3–5	
Zn	171	197	188	175	15–37	
Se	1.73	2.2	2.1	1.06	0.15–0.25	
D. Fatty acids (% of total fatty acids)	
16:0	17.6	17.2	17.2	15.4	-	
18:0	3.8	3.7	3.3	2	-	
18:1n−9	11.0	11.0	9.5	11.7	-	
18:2n−6	8.8	9.4	5.5	9	-	
20:4n−6 ARA	1.2	1.1	0.6	0.6	-	
20:5n−3 EPA	13.5	12.9	17.1	8.5	-	
22:6n−3 DHA	12.3	12.2	8.9	10.2	-	
DHA:EPA	0.91	0.94	0.52	1.20	-	
ARA:EPA	0.09	0.09	0.04	0.07	-	
Notes.

The juveniles were fed for at least three months and the broodstock for more than one year with the respective diets before sampling. The general requirements for fish according to NRC (2011) are given for comparison.

a Labrus feed is an ongrowing diet produced for Ballan wrasse by Skretting AS, Stavanger, Norway.

b Vitalis is a broodstock diet produced for marine fish by Skretting AS, Stavanger, Norway. It was blended with shrimp (Vitalis 75%, Shrimp 25%) to produce a moist diet at the rearing facility.

c Menadione bisulfite is a synthetic form of Vitamin K that is usually added to fish feed. Vitamin K is the sum of Phylloquinone and Menakinones 4-11. NR = not required; NT = not tested; = not given by NRC (2011).

Fish and sampling

Fish for the study on macronutrient composition were obtained from the production line at Marine Harvest Labrus AS (MHL). The initial weight was 1.27 ± 0.19 g, total length 4.53 ± 0.21 cm and condition factor 1.36 ± 0.10 (mean ± SD, n = 47). The fish were held in 100 L tanks, at 50 fish per tank, with flow-through water (3 L min-1), pumped from 150 m depth and heated to 16°C. Oxygen concentration was 7.5–8.0 mg L-1. The light regime was continuous light from fluorescent lamps placed above the tanks. The fish were fed continuously, using belt feeders at 7–10 g feed per tank per day. At the end of the study, the total length and weight of the fish were measured, 6 fish per tank were pooled for analyses of whole body macronutrient composition while livers from 25 fish per tank were dissected out, pooled and analyzed for macronutrient composition.

Three groups of cultured juvenile fish (mean weight 3–13 g, Table 6) were sampled for analyses of whole body nutritional composition from the production line of MHL. The fish were held in 25 m3 tanks with continuous fluorescent light, temperature 12–14°C, oxygen 7.5–8.0 mg L-1 and had been fed the diets listed in Table 3 for more than three months. Wild fish (430 ± 138 g, mean ± SD, n = 10, Table 6) were caught in August 2011 in fish traps near Austevoll Aquaculture Research Station which is situated near Bergen in Western Norway. The size of the wild fish enabled analyses of all nutrients in individual fish, but the size difference between wild and cultured fish was not optimal. However, it was not feasible to obtain wild fish in the size range of the available cultured fish within the time and resource limits of the project. Samples of 30–50 pooled cultured fish or individual wild fish were killed with an overdose of MS222, immersed in fresh water, frozen on dry ice or at −80°C, transferred to NIFES on dry ice, thawed and homogenized using a kitchen blender. Sample portions were frozen and stored at −80°C for future analyses of all analytes except minerals. Samples for analysis of minerals were freeze dried and further homogenized by grinding on a Retsch Mill (Retsch Gmbh, Haan, Germany).

Table 3 Analytical methods for the different nutrients.

Analyte	Principle	Reference	
Dry matter	Gravimetric after freeze drying	Hamre & Mangor-Jensen (2006)	
Protein	N × 6.25, Leco N analyzer	Hamre & Mangor-Jensen (2006)	
Taurine	Total amino acids	Espe et al. (2006)	
Lipid (tissues)	Gravimetric after ethyl solven extraction	Lie, Waagbø & Sandnes (1988)	
Lipid (feed)	Gravimetric after acid hydrolysis	EU directive 84/4 1983	
Fatty acids	Transmethylation extraction and GC/FID	Lie & Lambertsen (1991)	
Vitamin C	HPLC	Mæland & Waagbø (1998)	
Biotin	Microbiological	Mæland et al. (2000)	
Folic acid	Microbiological	Mæland et al. (2000)	
Niacin	Microbiological	Mæland et al. (2000)	
Pantothenic acid	Microbiological	Mæland et al. (2000)	
Vitamin B6	HPLC	CEN (2003a)	
Thiamine	HPLC	CEN (2003b)	
Riboflavin	HPLC	CEN (2005)	
Cobalamin	Microbiological	Mæland et al. (2000)	
Vitamin A	HPLC	Moren, Næss & Hamre (2002)	
Vitamin D	HPLC	CEN (1999)	
Vitamin E	HPLC	Hamre, Kolås & Sandnes (2010)	
Sum Vitamin K3	HPLC	CEN (2003c)	
Astaxanthin	HPLC	Ørnsrud et al. (2004)	
Macrominerals	ICP-MS	Julshamn et al. (2007)	
Microminerals	ICP-MS	Julshamn et al. (2004)	
Iodine	ICP-MS	Julshamn, Dahl & Eckhoff (2001)	

Captive broodstock (718 ± 91 g, mean ± SD, n = 14) for analyses of the nutrient composition of female gonads were also sampled from the production line of MHL. The fish had been held in 40 m3 tanks at a density of 5 kg m3 for more than one year and fed the moist diet consisting of 75% Vitalis (Skretting AS, Stavanger, Norway) and 25% shrimp. Water temperature was 8–10°C, oxygen 7.5–8.0 mg L-1, the light regime was set for delayed spawning and the tanks contained shelters made of plastic sheets where the fish could hide. The fish were scheduled to spawn in December and were sampled on the 14.12.11. The wild fish (521 ± 155 g, mean ± SD, n = 17) were obtained from a local fisherman and moved to MHL, where they were held for less than 1 week without feeding and sampled on the 5.05.10. The spawning season for Ballan wrasse in the wild lasts from May until July. If the gonad from one fish was too small to give material to all the analyses in a series, gonad from two fish were sampled for that series. The fish were killed by a blow to the head, then weight, total length, gonad weight and liver weight were measured (Table 7), the gonad was homogenized in a kitchen blender and portions of sample were distributed to tubes for the different analyses. The samples were immediately frozen on dry ice and transported to NIFES, where they were stored at −80°C until analyses.

Experimental design of the macronutrient study

The macronutrient experiment was conducted with a three-component mixture design (Cornell, 2011). This design allows a variation of protein, lipid and carbohydrate simultaneously, continuously and systematically, within given limits. Using 13 different diets (i.e., treatments), of which 12 diets were administered to fish in single tanks, it was possible to cover a wide range of nutrient compositions (Table 1B). One diet (the centre point treatment) was fed to fish in 3 tanks to obtain a measure of tank variation. The experiment lasted for 56 days.

Analytical methods

The nutrient composition of fish and diets were measured by routine methods established at NIFES. Table 3 gives an overview over the methods with references.

Calculations and statistics

Data are given as mean ± SD and differences and effects were considered significant at p < 0.05. Models describing the effects of macronutrient composition on weight, length, condition factor and nutrient composition of whole body were calculated using the software Design Expert v 8.0.4. (Stat-Ease Inc., MN, USA). Different models were fitted to the data and the recommended model with the best fit was chosen. Insignificant terms were removed when allowed by hierarchy rules.

Data on micronutrient composition of wrasse female gonad and juvenile whole body were analyzed using Statistica v 11 (StatSoft Inc., Tulsa, OK, USA). Data on gonads were first subjected to regression analyses to investigate possible relations between nutrient status and the gonadosomatic index (GSI). Gonads that weighed less than 5% of body weight (GSI’s less than 5) turned out to have a different nutrient status than the larger gonads (GSI at or above 5, Fig. 2) and were omitted from the study. The omitted fish all belonged to the farmed group. The data were then analyzed using the Student’s t-test and the Mann-Whitney U test.

Data on nutrient composition of juveniles were analyzed for homogenous variances using Levene’s test. Those nutrients where the data had homogenous variances were analyzed by one way ANOVA and Tukeys Honest Significant Difference post hoc test for unequal sample sizes. All juvenile nutrient status data were also analyzed using the Kruskal-Wallis test and nonparametric analyses of differences between groups were performed with the Mann-Whitney U test.

Results

The macronutrient experiment

The analyzed levels of macronutrients were different from the formulated levels. Analyzed protein was 3.5% higher than the formulated level at 77% supplementation but the difference decreased gradually to approximately 2% at the lower supplementation levels. Carbohydrate level in the diet was calculated by subtraction and was up to 4% lower than formulated in the 5% supplementation, but the difference decreased with increased supplementation and was close to zero at 20–25% supplementation. Analyzed ash was 0.7 ± 0.2% lower than formulated while analyzed and formulated lipid levels were similar. The weight percentage of raw materials contributing with dietary carbohydrate was 1.17–1.18 times the formulated carbohydrate levels in all cases. Further results are based on the formulated levels of macronutrients.

The variation in final weight of fish fed diets with systematic variation in protein, lipid and carbohydrate (Fig. 1A) followed the cubic model given in Table 4 (R2 = 0.96, p = 4∗10−4). The composition which gave the maximum final weight was approximately 70% protein, 10% lipid and 12.5% carbohydrate. The maximum and minimum measured final weights were 5.0 g and 3.3 g, respectively. Fish fed the commercial control diet weighed 3.4 g. A cubic model also gave the best fit to the data on total length (R2 = 0.84, p = 0.03; Fig. 1B; Table 4), the diet for obtaining the maximum length was 65% protein, 12% lipid and 16% carbohydrate. Maximum and minimum measured final lengths were 6.4 and 5.7 cm, respectively, while the fish fed the commercial control diet were 5.9 cm. For condition factor, a quadratic model gave the best fit (R2 = 0.86, p = 0.002; Fig. 1C, Table 4), with the highest factor at or above 78% protein, 10% or less lipid and 5% or less carbohydrate. The maximum and minimum measured condition factors were 1.85 and 1.52, respectively, while fish fed the control diet had a condition factor of 1.53. The variation in the other measured responses was not systematic, and could not be described by a significant model. The average values for all experimental groups in survival, hepatosomatic index and biochemical composition of whole body and liver are given in Table 5, together with the values for fish fed the commercial control diet.

Figure 1 Growth responses of Ballan wrasse fed varying dietary levels of macronutrients.

(A) Final weight. (B) Final length. (C) Final condition factor. The triangle represents the response surface for all possible combinations of protein, lipid and carbohydrate and the graded response represents variation in weight, length and condition factor where red is maximum and blue is minimum. The red dots represent the composition of the different diets.

Table 4 Models fitted to three component mixture design data investigating the optimum composition of dietary protein (P), lipid (L) and carbohydrate (CH) for Ballan wrasse.

The coefficients for the different terms and the significance of the terms are given for the different models. Cubic models were the basis for the models for length and weight and a quadratic model was used for condition factor. Insignificant terms were removed when appropriate.

	Weight	Length	Condition factor	
	Coefficient	P term	Coefficient	P term	Coefficient	P term	
P	0.026		0.050		0.023		
L	1.31	0.0003	0.98	0.0138	0.055	0.002	
CH	−1.26		−1.03		−0.028		
PxL	−0.024	0.0002	−0.017	0.019	−0.00078	0.003	
PxCH	0.019	0.0009	0.018	0.0397	0.00056	0.02	
LxCH	0.0060	0.19	0.0050	0.4281			
PxLxCH							
PxLx(P-L)	0.00012	0.004	0.00011	0.0454			
PxCHx(P-CH)			−1.7 × 10−6	0.169			
LxCHx(L-CH)	−0.00051	0.0009	−0.00027	0.0175			
Adjusted R2	0.96		0.84		0.86		
P model	0.0004		0.03		0.002		

Table 5 Responses without systematic variation in experiment 1; optimal composition of macronutrients for Ballan wrasse.

%	Average exp groups	Commercial control	
Survival	80 ± 7	96	
HSI	1.6 ± 0.2	1.2	
Whole body *	
Dry matter	22 ± 1	23	
Protein	71 ± 3	69	
Lipid	19 ± 2	20	
Glycogen	0.58 ± 0.26	0.83	
Ash	11 ± 1	10	
Liver *	
Dry matter	34 ± 3	35	
Protein	36 ± 5	37	
Glycogen	7.7 ± 3.2	9.7	
Total fatty acids	46 ± 11	34	
Notes.

* Macronutrients are given as % of DW.

Nutrient composition of the diets

Two diets fed to Ballan wrasse were sampled for analyses of nutrient composition: the Labrus diet and the Vitalis + shrimp diet. The Vitalis + shrimp diet was analysed in 2010 and 2011 to cover the period of feeding that may have affected the gonad composition in the broodstock. The results from analysis of the diets are given in Table 2. The Labrus diet was characterized by a low lipid level of 13% and a moderate protein level of 54%. The Vitalis diet and the diet blend with Vitalis and shrimp had high protein levels at 58 and 62% and lipid levels at 18 and 21%, respectively. Protein requirements in fish in general vary from 30 to 60% of dry matter (NRC, 2011). Taurine was only analyzed in the Labrus diet and in the Vitalis + shrimp diet in 2011 with similar results at 3.9 and 3.7 g kg-1 DM, respectively.

All the water-soluble vitamins were present above the requirements given by NRC (2011) in all diets. This was also true for Vitamin A and for Vitamin D in the diets where it was measured. Vitamin E was not analyzed in the Labrus diet, it was 10-fold above the requirement in the Vitalis diet, more than 5-fold above the requirement in the Vitalis + shrimp diet in 2010, but below the requirement in the Vitalis + shrimp diet in 2011. For Vitamin K both menadione bisulfite (MSB, the synthetic form used for feed supplementation) and the sum of Vitamin K originating from the feed ingredients were analyzed. The latter was slightly above 0.1 mg kg-1 in all diets, e.g., up to 10-fold higher than MSB. The requirement for Vitamin K in fish is not known. Astaxanthin was present in all diets at 36–57 mg kg-1, except in the Vitalis + shrimp diet in 2011 where astaxanthin was not detected. Astaxanthin from shrimp is probably largely present as esters, and is not detected by our analytical method. It is not known if Ballan wrasse has a requirement for astaxanthin.

The macrominerals, Fe and I were only analyzed in the Labrus diet and in the Vitalis + shrimp diet in 2011. Ca, Na and K were all present in the diets. Requirements are not given for these minerals by NRC (2011), which are probably obtained via sea water. The diets were all above requirements in concentrations of Mg, P and I, but the concentrations of Fe were below or in the very low range of requirements in fish (NRC, 2011). For the rest of the microminerals; Mn, Cu, Zn, Se, the levels in all diets were several-fold higher than requirements.

The fatty acid compositions of the diets were characterized by differences in the essential fatty acids. The Vitalis diet in 2010 had more than 1% ARA, 13% EPA and 12% DHA of total fatty acids (TFA), while the Vitalis diet in 2011 had 0.6% ARA, 17% EPA and 9% DHA. The Labrus diet had 0.6% ARA, 9% EPA and 10% DHA. The differences in essential fatty acids affected the fatty acid ratios. DHA:EPA was approximately 1 in the Labrus diet and the Vitalis + shrimp diet in 2011 and 0.5 in the Vitalis + shrimp diet from 2010. The ARA:EPA ratio was also lower in the Vitalis + shrimp diet from 2010 (0.04) than in the other diets (0.07–0.09).

Whole body nutrient profile of farmed and wild fish

Dry matter percentage increased with fish size from 19.7 to 25.6% of wet weight in fish with an average weight of 3.0 and 12.7 g, respectively (p(ANOVA) = 0.0009; p(Kruskal-Wallis test) = 0.015 (this sequence of tests is used throughout); Table 6). The dry matter of the largest cultured fish group was not different from the wild fish, which had an average weight of 430 g and a dry matter of 26.3%. Protein and glycogen measured on a dry matter basis did not differ between the analyzed groups. Fish fed Vitalis + shrimp had a higher lipid level than fish fed the Labrus diet (p(ANOVA) = 0.002, p(Kruskal-Wallis); ns), while the wild fish was intermediate in lipid. Taurine concentration was lower in fish fed the Vitalis + shrimp diet than in the Labrus fed fish and the wild fish.

Table 6 Nutrient composition (in dry matter) of whole body of cultured Ballan wrasse juveniles compared to wild caught Ballan wrasse.

Fish origin1		G02-20112	G01-20112	G01-02-20102	Wild fish3	P	P	
Feed		Labrus4	Labrus4	Vitalis + shrimp5	–	ANOVA	Kruskal Wallis	
A. Macronutrients and taurine	
Dry matter	g 100 g-1	19.7 ± 0.5a	22.3 ± 1.1ab	25.6 ± 0.3b	26.3 ± 2.4b	0.0009	0.015	
Protein	g 100 g-1	75.1 ± 1.3	73 ± 1	68 ± 4	71.1 ± 3.9	ns	ns	
Taurine	g kg-1	12.9 ± 1.2a	13.2 ± 1.2a	7.9 ± 0.2b	11.4 ± 0.9a	0.0001	0.019	
Lipid	g 100 g-1	6.9 ± 0.4a	9.4 ± 1.9a	22 ± 1b	12.8 ± 5.7ab	0.002	ns	
Glycogen	g 100 g-1	0.5 ± 0.1	0.6 ± 0.1	0.5 ± 0.1	0.7 ± 0.3	ns	ns	
B. Vitamins	
Vitamin C	mg kg-1	61 ± 4a	32 ± 4b	18 ± 2bc	9 ± 6c	< 10-6	0.004	
Biotin	µ g kg-1	327 ± 15a	270 ± 23b	203 ± 9c	178 ± 24c	< 10-6	0.005	
Folic acid	mg kg-1	1.1 ± 0.2a	0.9 ± 0.1ab	0.5 ± 0.0bc	0.3 ± 0.1c	10-6	0.005	
Niacin	mg kg-1	64 ± 7	51 ± 1	36 ± 1	47 ± 6	na	0.01	
Pantothenic acid	mg kg-1	20.5 ± 0.7a	14.9 ± 1.0b	12.1 ± 0.4bc	9.9 ± 1.1c	< 10-6	0.004	
Vitamin B6	mg kg-1	216 ± 25a	162 ± 33a	65 ± 6b	60 ± 22b	< 10-5	0.008	
Thiamine	mg kg-1	7.8 ± 0.6a	5.5 ± 0.5a	5.1 ± 0.9ab	2.6 ± 1.2b	10-4	0.004	
Riboflavin	mg kg-1	9.4 ± 0.3	9.2 ± 2.4	7.4 ± 1.4	4.9 ± 1.1	na	0.005	
Cobalamin	µ g kg-1	266 ± 30a	213 ± 29ab	147 ± 4bc	124 ± 24c	10-5	0.006	
Sum Vitamin A	mg kg-1	0.9 ± 0.2a	1.7 ± 0.8a	2.1 ± 0.1ab	17 ± 8b	0.002	0.003	
Vitamin D	mg kg-1	0.25 ± 0.09a	0.11 ± 0.03a	0.10 ± 0.03a	0.94 ± 0.19b	< 10-5	0.003	
Vitamin E6	mg kg-1	45 ± 13a	65	26 ± 0ab	18 ± 10b	0.001	0.04	
Sum Vitamin K	µ g kg-1	46 ± 8	18 ± 11	13 ± 0	81 ± 49	na	0.01	
C. Minerals	
Macrominerals (g kg-1)	
Ca		38 ± 3	43 ± 1	30 ± 5	41 ± 12	ns	ns	
K		15 ± 1	11 ± 3	12 ± 1	11 ± 1	na	0.04	
Mg		2.3 ± 0.3a	2.0 ± 0.2ab	1.4 ± 0.0c	1.8 ± 0.2bc	0.0002	0.01	
Na		10.8 ± 1.7	8.0 ± 1.2	5.6 ± 0.5	4.6 ± 0.5	na	0.004	
P		33 ± 6	32 ± 4	23 ± 5	29 ± 7	ns	ns	
Microminerals (mg kg-1)	
Fe		56 ± 7	52 ± 13	49 ± 14	90 ± 24	0.015	0.018	
Cu		2.6 ± 0.1	2.2 ± 0.2	1.9 ± 0.2	2.3 ± 0.4	ns	ns	
Mn		22.4 ± 2.0	22.5 ± 2.1	18.2 ± 1.3	24 ± 6	ns	ns	
Se		1.2 ± 0.0	1.2 ± 0.1	1.3 ± 0.1	1.4 ± 0.2	ns	ns	
Zn		86 ± 2a	73 ± 5ab	62 ± 4bc	57 ± 10c	0.0005	0.001	
D. Fatty acids (% of total fatty acids)	
16:0		16.5 ± 0.1	16.3 ± 0.1	14.1 ± 0.1	15 ± 1	ns	ns	
18:1n−9		13.4 ± 0.1	14.0 ± 0.4	13.2 ± 0.3	15 ± 2	ns	ns	
18:2n−6		10.4 ± 0.2b	11.9 ± 0.6a	6.8 ± 0.1c	1.5 ± 0.2d	< 10-2	0.003	
20:4n−6 ARA		0.6 ± 0.1	0.5 ± 0.0	0.7 ± 0.1	2.4 ± 0.9	na	0.004	
20:5n−3 EPA		8.4 ± 0.1a	7.5 ± 0.4a	15.1 ± 0.1b	9.0 ± 1.1a	10-6	0.02	
22:6n−3 DHA		19.9 ± 0.5	16.2 ± 1.9	10.9 ± 0.0	15.9 ± 3.1	na	0.03	
DHA:EPA		2.36 ± 0.05a	2.13 ± 0.16a	0.72 ± 0.00b	1.81 ± 0.43a	0.001	0.02	
ARA:EPA		0.07 ± 0.01	0.07 ± 0.00	0.05 ± 0.01	0.27 ± 0.13	na	0.005	
Notes.

ANOVA analyses were conducted for data with homogenous variances. Different letters in superscripts indicate significant differences, p < 0.05. The Kruskal-Wallis test was performed on all data. Differences were considered significant at p < 0.05. ns = not significant. na = not analysed due to variances not being homogenous.

1 G. Generation; fish from MHL. Marine Harvest Labrus.

2 The samples were 30–50 pooled fish per tank.

3 The wild fish were caught in fish traps near Austevoll Aquaculture Research Centre in Western Norway. 10 individual fish were analyzed.

4 Labrus feed is an ongrowing diet produced for Ballan wrasse by Skretting AS. Stavanger. Norway. The feed had been fed for 3 months and 4–5 months before sampling.

5 Vitalis is a broodstock diet produced for marine fish by Skretting AS, Stavanger, Norway. It was blended with shrimp at the rearing facility. The feed had been fed for 4–5 months before sampling

6 Sufficient material to analyze one replicate only in G01-2011.

The concentrations of water-soluble vitamins were similar or higher in cultured than in wild fish (p ≤ 10−4; p ≤ 0.01; Table 6). The exception was niacin, which was lower in fish fed the Vitalis + shrimp diet than in wild fish (p(MwU test) = 0.04). The concentrations of Vitamin A and Vitamin D were up to 9 fold higher in wild fish than in cultured fish (p ≤ 0.03; p = 0.003; Table 6). The concentration of Vitamin E was similar or higher in cultured fish than in wild fish. The concentration of Vitamin K was lower in the groups G01-2011 (Labrus feed) and G01-G02-2010 (Vitalis + shrimp) than in wild fish (p ≤ 0.04, MwU-test) and similar to wild fish in group G02-2010 (Labrus feed).

Of the macrominerals (Table 6), there was no difference between wild and cultured fish in Ca and P. However, the standard deviations were large and may have masked a possible difference between the fish fed the Vitalis + shrimp diet and the wild fish which had average Ca concentrations of 30 and 41 g kg-1 DM and P concentrations of 23 and 29 g kg-1 DM, respectively. Furthermore, there was a good correlation between Ca and P concentration in these samples (R2 = 0.82). In the cultured fish, Mg concentrations appeared to decrease with increased fish weight (p = 0.0002; p = 0.01). Mg concentration in G02-2011 was significantly higher than the concentration in G01-02-2010 and in wild fish (p = 0.0005 and p = 0.003, respectively). The other cultured groups had intermediate concentrations. K was higher in G02-2011 than in wild fish (p = 0.014), but similar to wild fish in the other cultured groups. Na concentration in the cultured fish decreased with increasing size (p(KW test) = 0.004) and was similar to the wild fish in the largest group (G01-02-2010).

Of the microminerals, iron was lower in cultured than in wild fish (p = 0.02; p = 0.02; Table 6). Zinc decreased with increasing fish weight (p = 0.0005; p = 0.001) and was similar in the largest cultured group and in wild fish. There were no differences between wild and cultured fish in the concentrations of Cu, Mn and Se.

There were no differences in the levels of the fatty acids 16:0 and 18:1n−9 between the fish groups (Table 6). The percent 18:2n−6 of total fatty acids (TFA) varied between the groups (p = 0.01; p = 0.003). Even though there were small differences in 18:2n−6 between G01 and G02-2011, it was nominally small and the groups can be regarded as biologically similar. The concentration in G01-02-2010 was significantly different from all other groups and intermediate, while the wild fish had the lowest concentration. Arachidonic acid (ARA; 20:4n−6) was similar in the three groups of cultured fish at 0.5–0.7% of TFA while the average concentration in wild fish was 2.4% of TFA (p(KW test) = 0.004). EPA was higher in the fish fed the Vitalis + shrimp diet than in the other groups (p = 10−6; 0.02) which were similar. DHA was lower in fish fed the Vitalis + shrimp diet (p(MW U test) = 0.04) than in wild fish, but similar to wild fish in the other cultured groups. The DHA/EPA ratio was the lowest at 0.72 in the fish fed the Vitalis + shrimp diet, and statistically similar at average values of 1.8–2.4 in the other groups (p = 0.001; p = 0.02). The ARA/EPA ratio was higher in the wild fish (average 0.17) than in the cultured fish (average 0.05–0.07; p(KW test) = 0.005).

Nutrient profiles of female gonads from mature wild wrasse and mature wrasse held in captivity for one year

There were no significant differences in weight or length of the wild and captive fish sampled for analyses (Table 8), however, the captive fish had higher mean values, and a large variation may have masked possible differences. The gonadosomatic index (GSI) was significantly lower in captive than in wild fish (p(t-test) = 0.01). The concentration of nutrients in the gonads changed when GSI dropped below 5, as shown in Fig. 2. Therefore, fish with GSI lower than 5 were omitted from the data on nutrient composition of female gonad given in Table 8.

Figure 2 Variation in nutrient levels with increasing gonadosomatic index (GSI) in broodstock.

Filled squares, captive fish; filled circles, wild fish.

Table 7 Body weight and length, condition factor (CF), gonadosomatic index (GSI) and hepatosomatic index (HSI) of captive and wild Ballan wrasse females sampled for analyses of nutrient composition of gonads.

Fish	Weight (g)	Total length (cm)	CF	GSI	HSI	
Captive	718 ± 91	35 ± 1	1.72 ± 0.13	4.5 ± 2.5a	2.08 ± 0.40	
Wild	521 ± 155	31 ± 3	1.69 ± 0.29	8.4 ± 2.3b	1.95 ± 0.40	
Notes.

Different letters in superscripts indicate significant differences (t-test, p = 0.01).

Table 8 Nutrient composition (in dry matter) of female gonads from captive and wild Ballan wrasse.

The captive wrasse had been held at Marine Harvest Labrus for at least one year and fed a moist diet consisting of 75% Vitalis and 25% shrimp. Ns = not significant, na = not analysed.

		Captive	Wild	P	P	
N		6	10	t-test	MW-u test	
A. Macronutrients and taurine	
GSI		7.0 ± 1.9	8.7 ± 2.7	ns	ns	
Dry matter	g 100 g-1	20.3 ± 2.4	21.0 ± 1.0	ns	ns	
Protein	g 100 g-1	76 ± 0	75 ± 1	ns	ns	
Taurine	g kg-1	5.6 ± 1.8	10.4 ± 1.8	0.002	0.014	
Total fatty acids	g 100 g-1	9.3 ± 1.4	10.2 ± 1.4	ns	ns	
B. Vitamins (mg kg-1 DM)	
Vitamin C		190 ± 37	140 ± 67	ns	ns	
Biotin		0.99 ± 0.29	0.86 ± 0.11	ns	ns	
Folic acid		2.2 ± 1.0	2.0 ± 1.0	ns	ns	
Niacin		64 ± 7	49 ± 7	0.001	0.01	
Pantothenic acid		64 ± 22	43 ± 7	0.01	0.03	
Vitamin B6		20 ± 5	12 ± 1	0.00007	0.006	
Thiamine		14 ± 5	3.7 ± 1.9	0.00003	0.002	
Riboflavin		17 ± 6	16 ± 3	ns	ns	
Vitamin D		0.31 ± 0.23	0.66 ± 0.33	0.04	0.04	
Vitamin E		387 ± 127	211 ± 49	0.0007	0.002	
Vitamin K		na	0.058 ± 0.027			
Astaxanthin		0.8 ± 0.2	0.0 ± 0.0	ns	0.0014	
C. Minerals	
Macrominerals (g kg-1 DM)	
Ca		0.47 ± 0.25	0.33 ± 0.05	ns	ns	
Na		7.9 ± 2.2	6.0 ± 0.5	0.01	0.03	
K		15.9 ± 3.5	12.8 ± 1.5	0.025	ns	
Mg		0.73 ± 0.19	0.58 ± 0.11	ns	ns	
P		12.8 ± 1.5	11.9 ± 0.9	ns	ns	
Microminerals (mg kg-1 DM)	
Fe		62 ± 12	43 ± 9	0.003	0.004	
I		0.61 ± 0.17	1.48 ± 0.58	0.003	0.004	
Mn		5.4 ± 1.5	4.1 ± 1.0	ns	0.03	
Cu		4.8 ± 1.4	4.8 ± 0.8	ns	ns	
Zn		262 ± 47	169 ± 24	0.0001	0.001	
Se		2.8 ± 0.3	3.2 ± 0.8	ns	ns	
D. Fatty acid (% of total fatty acids)	
16:0		22 ± 1	22 ± 1	ns	ns	
18:0		5 ± 1	5 ± 0	ns	ns	
18:1n−9		7 ± 0	8 ± 1	0.001	0.003	
20:4n−6 ARA		2 ± 0	6 ± 2	0.00002	0.001	
20:5n−3 EPA		17 ± 1	12 ± 1	< 10-7	0.001	
22:6n−3 DHA		25 ± 1	30 ± 4	0.01	0.015	
DHA:EPA		1.5 ± 0.1	2.5 ± 0.5	0.0004	0.0014	
ARA:EPA		0.10 ± 0.03	0.49 ± 0.13	< 10-5	0.0014	

GSI was similar in the two groups after fish with low GSIs had been removed from the group of captive fish (Table 8). Dry matter, protein and total fatty acids were also similar in gonads from wild and captive fish, but taurine was lower in captive than in wild gonad (p(t-test) = 0.002; p(MwU test) = 0.014 (this sequence of tests is used throughout); Table 8).

The water-soluble vitamins and Vitamin E were similar or higher (p ≤ 0.01; p ≤ 0.03) in the captive group, compared to the wild group while the concentration of Vitamin D was approximately half in the captive fish compared to wild fish (p = 0.04; p = 0.04). Vitamin K was not analyzed in the captive group due to a shortage of sample material. Astaxanthin was not detected in the gonads from wild fish, but was present in the gonads from captive fish.

Of the minerals, no differences were found between wild and captive fish in Ca, Mg, P, Cu and Se. Na (p = 0.01; p = 0.03) and K (p = 0.025; ns) were higher in captive than in wild fish. Moreover, a higher concentration of Mn (30%; ns; p = 0.03) and Zn (50%; p = 0.0001; p = 0.001) was determined in captive fish compared to wild fish, while I was present in captive fish at about 1/3 of the concentration in wild fish.

The most important differences in fatty acid composition were those of ARA, EPA and DHA. ARA was lower (p = 0.0002; p = 0.001), EPA higher (p < 10−7; p = 0.01) and DHA lower (p = 0.01; p = 0.015) in gonads from captive compared to wild fish. This resulted in large differences in essential fatty acid ratios, DHA:EPA being 1.53 and 2.53 (P = 0.0004; p = 0.0014) and ARA:EPA ratios being 0.10 and 0.49 (p < 10−5; p = 0.0014) in gonads from captive and wild fish, respectively. There was also a statistically significant difference in 18:1n−9 which amounted to 7 and 8% of TFA, respectively (p = 0.001; p = 0.003).

Discussion

The present study aimed at developing functioning diet formulations for Ballan wrasse within a short timeframe. Currently, the industry is establishing wrasse culture for controlling salmon lice, but fulfilling the nutritional requirements of both the ongrowing and broodstock fish are bottlenecks in the production line. To determine the optimum macronutrient composition in diets for juveniles we have run a three component mixture design varying protein, lipid and carbohydrate in a systematic manner and recording differences in growth, survival and body composition. One can assume that broodstock have similar macronutrient requirements as juveniles, however, this should be validated in future studies. To investigate whether fish in culture obtain a sufficient supply of taurine, fatty acids, and micronutrients, we analyzed the status of these nutrients in cultured and wild fish. Wild fish were used as a reference on the assumption their nutrient requirements are covered by their natural diet. Whole body analyses were used to assess the nutrient status of juveniles and analyses of female gonads to investigate broodstock nutritional status.

In the experiment on dietary macronutrient composition, the best growth (length and weight based) was obtained with diets containing 65–70% protein, 10–12% lipid and 12.5–16% carbohydrate. There was a slight difference in maximums between the weight and length data; the highest final weight was obtained at slightly higher protein and lipid levels and a lower carbohydrate level than the highest final length. This may have been caused by overfeeding in fish fed the diets with high protein contents. Until now, Ballan wrasse has had a satisfactory feed intake only when the feed is supplemented with shrimp or shrimp meal (A Nordgreen, E Grøtan and K Hamre, unpublished data; I Opstad, PG Kvenseth, P Jensen and AB Skiftesvik, unpublished data). In the present study, the protein source was a blend of cod fillet and shrimp at a fixed ratio which may have led to higher feed intake in fish fed the diets high in protein, and therefore high in shrimp. Due to the small size of the feed particles, feed intakes were not measured. However, the increased condition factors in fish fed high protein diets indicates that feed intake was probably high in these fish groups. Based on these considerations, lengthwise growth may be a better indicator of optimum macronutrient composition than growth in weight and the optimum diet composition would then be 65% protein, 12% lipid and 16% carbohydrate. Fish fed such a diet would have a condition factor at approximately 1.7, which is similar to the average condition factor of the wild mature fish sampled for analyses of female gonads. The optimum diet for wrasse juveniles therefore contains relatively high levels of protein, carbohydrate and a low lipid level.

The lipid level in the diet was increased with the addition of pure oil in the form of triglycerides. Although all diets contained relatively high concentrations of phospholipids from both the marine feed ingredients and the addition of soy lecithin, there was still a large reduction in the relative phospholipid concentrations with increase in total lipid level. In a later experiment, we have shown that wrasse have higher growth rates when the added dietary lipid is in the form of phospholipids as compared to neutral lipids (Ø Sæle, A Nordgreen, AB Skiftesvik and K Hamre, unpublished data). Thus the optimum dietary macronutrient composition may be dependent on lipid quality.

The commercial control feed was included to normalize between experiments. It had been previously tested against other commercial feeds and proven to give better results (E Grøtan, unpublished data). It differed from the experimental diets in several aspects, but most importantly, it did not contain shrimp. This alone may explain the low growth and condition factor in fish fed the commercial control diet compared to those fed the best experimental diets.

In the experiments where the nutrient profiles in wild and cultured fish were analyzed, the cultured broodstock were fed on a diet blend with 75% Vitalis and 25% minced shrimp. It was a moist diet, blended and pelleted on site. The protein content was similar, at 65%, and the lipid level slightly higher, at 18%, than the optima found in the macronutrient experiment. The juveniles were fed either with the Vitalis + shrimp diet or with a diet developed for Ballan wrasse (Labrus, Skretting AS) which contained 54% protein and 13% lipid.

Fish whole body dry matter increased with fish size up to 12.5 g. Such a weight/dry matter relationship is a common trait for several fish species (Hamre et al., 2002). Dry matter did not differ in gonads from wild and captive fish. Lipid was higher in whole bodies of juveniles that had been fed the Vitalis + shrimp diet than in those fed the Labrus diet. This may have been caused by a higher dietary lipid level, which often increased body lipid content in fish (Aksnes, Hjertnes & Opstvedt, 1996; Hamre et al., 2003; Karlsen et al., 2006). However, the Vitalis + shrimp diet was moist and included fresh shrimps, which may have stimulated increased feed – and hence energy intake, contributing to increased body lipid levels. There were no differences in lipid levels between fish fed the Vitalis + shrimp diet and the wild fish, either for juveniles or for broodstock. Protein in whole body given on a dry weight basis was similar among all groups of juveniles and did not differ in wild and captive fish female gonads. Protein levels are largely determined by the genetic code of healthy and feeding animals, however, when measured on a dry weight basis it may be lowered at high levels of body lipid, since lipid replaces water in the body. There was a tendency to lower protein level in the Vitalis + shrimp group, but there was no significant difference. Taurine is a non protein amino acid which, among others, is involved in osmoregulation and bile salt production. It is essential for cats and has been shown to be essential in the early stages of some marine fish species (Sturman, 1993; Chen et al., 2005; Chen et al., 2004; Pinto et al., 2010). The level of taurine was lower both in juveniles and in broodstock fish fed the Vitalis + shrimp diet compared to wild fish and fish fed the Labrus diet. The taurine levels in the two diets were similar, so the difference was probably due to differences in absorption, utilization and/or excretion of taurine, which again may be coupled to differences in dietary lipid levels. The requirement for taurine needs further investigation.

The water-soluble vitamins consist of Vitamin C and the B-vitamins. Vitamin C functions as a water-soluble antioxidant and is also involved in numerous biochemical reactions, e.g., hydroxylation of proline to hydroxyproline during synthesis of connective tissue (Sandnes, Torrissen & Waagbø, 1992; Meister, 1994). The B-vitamins are cofactors in intermediary metabolism. The dietary concentrations of water-soluble vitamins were mainly at or above the requirements in fish given by NRC (2011) and similar or higher in cultured fish compared to wild fish, both for juveniles and broodstock. The one exception was the level of niacin in the whole bodies of juveniles fed the Vitalis + shrimp diet, which was lower than in wild fish. The very high levels of some of the water-soluble vitamins in the diets were not reflected in the fish, possibly because excess of water-soluble vitamins are easily excreted from the body.

The lipid soluble vitamins consist of Vitamins A, D, E and K. Vitamin E is a lipid soluble antioxidant protecting the fish against in vivo lipid oxidation (Hamre, 2011). Vitamin E was present in cultured fish at or above the levels found in wild fish. Vitamin A is involved in regulation of cell proliferation and differentiation, and among others, is important for the integrity of the skin and development of body and organ axes in embryogenesis (Maden, 1994). Astaxanthin is converted to Vitamin A in Atlantic halibut (Moren, Næss & Hamre, 2002), possibly also in other fish, and can be considered a pro-Vitamin A form. Vitamin A levels in the diets were slightly above the optimum level of 2.4 mg kg-1 dry diet found for Atlantic halibut by Moren et al. (2004), while the minimum fish requirement given by the NRC (2011) is 0.8 mg kg-1 dry diet. However, Vitamin A was on average 10 fold higher in wild fish than cultured fish, but the individual variation in the wild fish was high. Analysis of Vitamin A in female gonads requires a specialized method and was not performed in this study. The body level of lipid soluble vitamins often increases linearly with increasing supplementation to far above the requirement (Hamre et al., 1997; Moren et al., 2004) and lower concentrations of these vitamins in cultured than in wild fish does not necessarily indicate deficiency. The required Vitamin A status in Ballan wrasse is therefore not known and needs further study. Vitamin D is involved in regulation of calcium and phosphorus homeostasis, important, among other things, for bone health (Lock et al., 2010). Vitamin D was also higher in wild fish than in cultured fish, 4- to 9-fold fold in the juveniles and 2-fold in the broodstock. The Vitamin D status in the different cultured juvenile groups was similar and Vitamin D levels were similar in the two analyzed diets, 2–10 fold higher than the requirement given by NRC (2011). As for Vitamin A, the required Vitamin D status in Ballan wrasse is not known and should be investigated further. Vitamin K is involved in carboxylation reactions, such as blood clotting and bone formation (Krossøy, Waagbø & Ørnsrud, 2011). The dietary concentration of Vitamin K originating from the feed ingredients was several fold higher than the added menadione bisulfite (MSB) and similar in the different diets. The concentration of Vitamin K in the whole bodies of cultured juveniles seemed to decrease with increasing fish size. The two largest cultured juvenile groups had a Vitamin K status that was < 25% of the average wild fish status. Again, the wild fish data showed large variation. As for Vitamins A and D, the requirement of Vitamin K in Ballan wrasse needs further investigation.

Calcium, magnesium and phosphorus can be classified as bone minerals. In broodstock, there were no differences in the concentration of these minerals between wild and cultured fish. No significant difference was found in calcium and phosphorus between wild and cultured juveniles. Fish fed the Labrus diet had similar mean values as the wild fish, but in fish fed the Vitalis + shrimp diet, the mean values of calcium and phosphorus levels were about 75% of the levels found in wild fish. For phosphorus, this corresponded to a difference in dietary levels. Magnesium was significantly lower in fish fed the Vitalis + shrimp diet than in fish fed the Labrus diet. This corresponded to a difference in the dietary magnesium concentration, which was, however, 4- to 6-fold higher than the requirement given by NRC (2011). None of the cultured fish groups had significant lower magnesium status than the wild fish, but the mean level in fish fed the Vitalis + shrimp diet was again 75% of the average in wild fish, while Labrus feed gave similar or higher mean values than in wild fish. Therefore, even though the differences were not significant, the requirements for bone minerals should be further investigated.

Sodium and potassium are classified as electrolytes, involved in osmoregulation in extra and intracellular spaces, respectively. Both minerals decreased in concentration with increasing fish weight. Potassium was 50% and sodium 100% higher in the smallest cultured group of juveniles versus the wild fish, but no other differences in whole fish were observed. Both sodium and potassium were higher in gonads from captive compared to wild fish. The captive and cultured fish had a high frequency of wounds and fin erosion, which may have led to an influx of sodium from the sea water. Increased intracellular potassium may then have been required to osmoregulate against the high extracellular sodium. The problem with excess electrolytes can probably be solved only by identifying the causes for the wounds and implementing the necessary rearing adjustments. An extensive amount of work is currently invested in this issue.

In the present study, diet levels of iron were at or below the minimum requirement given by NRC (2011). Still, iron was higher in gonads from cultured fish than in wild fish. In juveniles, the iron status in the cultured fish groups was similar to each other, but lower than in wild fish. One could try to assess iron status in fish fed a diet supplemented with iron in the upper range of requirements and see if it becomes more similar to that in wild fish. Iodine was lower in gonads from cultured compared to wild fish, although dietary iodine concentration was well above the requirement given by NRC (2011). Zinc levels were higher in gonads from cultured compared to wild fish. The dietary levels of zinc were similar, while zinc in juveniles appeared to decrease with increasing fish size, the largest cultured group being similar to wild fish. It is not known if the ranges of zinc found in the present study are within the safe range for Ballan wrasse. There were no differences in selenium and copper between wild and cultured fish. Overall, the iron, zinc and iodine requirements in Ballan wrasse require further investigation.

The fatty acid composition was quite different between wild and cultured fish, and between fish fed the Vitalis + shrimp diet compared to fish fed the Labrus diet. ARA was 3- to 5-fold higher in wild versus cultured fish, both in broodstock and juveniles. EPA was higher in fish fed the Vitalis + shrimp diet, but similar in juveniles fed the Labrus diet and wild fish. DHA was slightly lower in juveniles fed the Vitalis + shrimp diets and in broodstock than in wild fish. Consequently, the DHA:EPA ratio was approximately 2 in fish fed the Labrus diet which is the recommended level in eggs (Sargent, 1995). Broodstock and juveniles fed the Vitalis + shrimp diet had DHA:EPA ratios of 1.5 and 0.72, respectively. The ARA:EPA ratio was < 0.1 in the cultured fish and 0.3–0.5 in wild fish. This ratio is hypothesized to be important for spawning performance in marine fish (Furuita et al., 2003; Mazorra et al., 2003). In the present study, the broodstock diets contained 0.6% ARA, whereas in cod a dietary level of ARA of 1–2% of fatty acids gave the best spawning performance (Norberg et al., 2009). Broodstock diets for Ballan wrasse should probably be supplemented with more ARA, while the ARA level in feed for ongrowing may be less critical. It is not known if the differences in EPA, DHA and their ratios are critical for Ballan wrasse. A number of marine fish species are fed on diets high in plant ingredients, without major negative effects. The safe ranges of diet fatty acid composition for Ballan wrasse should be investigated further.

Conclusion

The optimum dietary macronutrient composition for juvenile Ballan wrasse according to the present study is 65% protein, 12% lipid and 16% carbohydrate, based on maximum lengthwise growth. The high optimum protein content may have been affected by the quality of the dietary ingredients, since shrimp was used as a fixed part of the protein source and may have functioned as an attractant. Furthermore, later studies have shown that the lipid class composition of added lipid has a large effect on growth, and lipid quality will probably affect the optimum macronutrient composition.

Based on chemical analysis, the diets used for culture of Ballan wrasse contained sufficient amounts of the micronutrients, except iron, according to NRC (2011). The macronutrient and the fatty acid compositions were slightly different from assumed optima.

Of the nutrients analyzed in juveniles and broodstock, the one in least need of further investigation is the water-soluble vitamins, Vitamin E, selenium and copper. The high levels of electrolytes in cultured fish were probably due to wounds and fin erosion and are of minor interest when it comes to nutrition. The levels of bone minerals in wild and cultured fish were not significantly different, but should be investigated further as large differences were observed in mean values. Adjustments should be made in ARA and iodine for broodstock, and in iron both for broodstock and juveniles.

Vitamins A, D and K were all higher in wild fish than in cultured fish. These vitamins do not usually show a clear relationship between body level and requirements, so it is unknown if the requirements were covered in the cultured fish. Moreover, there are several different forms of each of these vitamins, and there are relatively few studies on the requirements in fish, at least for Vitamins D and K. These are all reasons that the requirements for these vitamins in Ballan wrasse deserve further studies. Taurine and zinc are other possible candidates for more in-depth investigations.

Many thanks to Kjersti Ask, NIFES, who organized the analyses of the samples, to the technical staff at NIFES who performed the analyses and to Karen Kvalheim and Henning Sandøy at Marine Harvest Labrus, Reidun Bjelland and Anne Berit Skiftesvik at the Institute of Marine Research for help with sampling of fish. We also thank Samuel J. Penglase for improving the language of the manuscript.

Additional Information and Declarations

Competing Interests

Author Contributions

Animal Ethics

Kristin Hamre is an Academic Editor for PeerJ. Espen Grøtan and Olav Breck are employed by Marine Harvest to work with the development of commercial culture of Ballan wrasse. Skretting AS produces the commercial diets fed to the captive fish analysed in the study.

Kristin Hamre and Andreas Nordgreen conceived and designed the experiments, performed the experiments, analyzed the data, contributed reagents/materials/analysis tools, wrote the paper.

Espen Grøtan conceived and designed the experiments, contributed reagents/materials/analysis tools, wrote the paper.

Olav Breck conceived and designed the experiments, performed the experiments, contributed reagents/materials/analysis tools, wrote the paper.

The following information was supplied relating to ethical approvals (i.e., approving body and any reference numbers):

This study was carried out within the Norwegian animal welfare act guidelines (code 750.000) at Marine Harvest Labrus. As the fish trials were assumed to be nutrition trials based on all available studies up to the date of the trial, no specific permit was required under the guidelines.

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
