# Peer review of "A holistic approach to development of diets for Ballan wrasse (Labrus berggylta) – a new species in aquaculture"

_PeerJ, doi:10.7717/peerj.99_

## Round 0.1 · original submission · Minor Revisions

Your approach to development of diets for Ballan wrasse is scientifically and methodologically sound, being an interesting an valid contribution to the fields fish nutrition and aquaculture.

I recommend however a thorough revision of the sentence construction and grammar.

Reviewer 1 ·

Basic reporting

The paper meet journal standards. It is well deisgned and written. No comments to the authors

Experimental design

Teh experimental design is correct. It is a shame that a not so high valued species has been considered for the study, The species selected is just cultured for delousing salmon and it is not produced for human consumption. However the approach can be used for other more interesting species (from a human consumption point of view) and both the experimental design, the tools and the results will be useful for other studies

Validity of the findings

The findings are really interesting, especially the approach (holistic and considering a lot of nutrients involved either in the diets or in the wild animals) perhaps too many "research is needed" for most of the nutrients analysed but considering the approach and the quantity of nutrients analysed it is logic to find no clear meaning of the requirement or the need in the fish

Additional comments

The paper is very, very interesting, especially for the high number of nutrients considered and analysed and for the approach given to the experiment. A more economically intersting species would have been better (i.e. salmon or halibut) from a scientific point of view but in any case the techniques, results and conclusions are also useful for other species

Reviewer 2 ·

Basic reporting

The basic reporting is of a good standard following scientific format and with appropriate statistical analyses.

However, the sentence construction and grammar require to be checked throughout and there are numerous typographical errors, e.g. reference list under NRC 2011, Table 2 first column, Table 8a title.

On the surface the ms appears too long but many of the pages have only a legend title.

It would have been easier to comment on specific areas if the lines had been numbered.

I am recommending "Minor revisions" solely because of typo errors and poor expression, sentence construction and grammar: the ms needs to be checked.

Experimental design

The design is very thorough looking at both juvenile and broodstock diets and the flesh composition of both wild and reared wrasse. The statistics meet reequirements.

Validity of the findings

The discussion and conclusions are valid and the data sets robust.

Page 5 authors indicate that it was not possible to get wild fish of similar size; is this a drawback? and does it limit the comparison with farmed wrasse.

Additional comments

Minor comment:
line 1 use the term sea lice rather than louse
Line 3 there may be examples of the development of resistance but there are no data to confirm that this is occurring in all fish farm areas.
Line 6 but environmental agencies monitor discharges
Page 3 assertion that ballan wrasse production established but numbers are still low and there are many technical challenges
Labrus diet:: was this used for both growers and broodstock?
p.21 diets high in plant ingredients: are authors saying that these will meet the nutritional requirements of wrasse?

---

## Round 0.2 · Minor Revisions

The revisions required are really minor. Two typos: 1) line 9 of abstract: word ANALYSED is missing after "were"; and 2) page 2, line 18: correct spelling of PROBLEM)

---

## Round 0.3 · accepted · Accept

I enjoyed the well structured novel approach for a quick screening of nutritional requirements of novel species.